

# QUADICA v2: Extending the large-sample data set for water QUAlity, DIscharge and Catchment Attributes in Germany

Pia Ebeling[1],
Alexander Hubig[1],
Alexander Wachholz[3],
Ulrike Scharfenberger[4],
Sarah Haug[1],
Tam Nguyen[1],
Fanny Sarrazin[5],
Masooma Batool[2],
Andreas Musolff[1],
Rohini Kumar[2]

[1]Department of Hydrogeology, Helmholtz Centre for Environmental Research-UFZ, Leipzig, 04318, Germany
[2]Department of Computational Hydrosystems, Helmholtz Centre for Environmental Research-UFZ, Leipzig, 04318, Germany
[3]Department of Inland Surface Waters, German Environment Agency-UBA, Dessau, 06844, Germany
[4]Department Aquatic Ecosystems Analysis and Management, Helmholtz Centre for Environmental Research-UFZ, Magdeburg, 39114, Germany
[5]Université Paris-Saclay, INRAE, UR HYCAR, 92160 Antony, France

*Correspondence to*: Pia Ebeling (pia.ebeling@ufz.de)

# Abstract

The QUADICA version 2 dataset significantly expands upon the first version of QUADICA (water QUAlity, DIscharge and Catchment Attributes for large-sample studies in Germany), by incorporating more recent data, additional water quality and driver variables, and more stations with concurrent water quantity data. Specifically, QUADICA v2 extends the water quality time series of the first version up to 2020 and introduces new variables, including water temperature, oxygen, and chlorophyll-a concentrations, as well as concentrations of ammonium, sulfate, and geogenic solutes like calcium. These additions enable a more comprehensive understanding of ecological impacts, including eutrophication effects, and water quality dynamics across catchments. Furthermore, the number of stations with both water quality and quantity data has effectively doubled – now covering 637 out of the total 1386 stations – by integrating QUADICA with the CAMELS-DE and Caravan-DE datasets. The inclusion of time series on point and diffuse sources of both nitrogen and phosphorus allows for more thorough investigations of driver-response relationships and nutrient export from catchments. To facilitate visualization and exploration of QUADICA, we provide a user-friendly, interactive R application alongside the online data repository, as well as a browser-based web app for inspecting the dataset. This makes QUADICA v2 a comprehensive dataset that spans from driver to impact variables, offering a valuable resource for researchers and practitioners.

# 1 Introduction

High water quality is critical for the health of aquatic ecosystems and humans. Understanding the spatial and temporal variability in water quality variables is essential for effective management and conservation of water resources. Observational data are the key to propelling our understanding of hydrological and biogeochemical processes and complex interactions. Large-sample hydrology (LSH) addresses the "need to balance depth and breadth" (Gupta et al., 2014) and has thus become a cornerstone to understand the generality of patterns and processes across diverse landscape and climate settings.

LSH data sets that combine stream observations with contextual data on catchment attributes and driving forces have gained momentum in recent years. For water quantity, the CAMELS data sets available in several countries (Addor et al., 2017; Alvarez-Garreton et al., 2018; Coxon et al., 2020; Chagas et al., 2020; Fowler et al., 2021; Loritz et al., 2024) and the globally consistent data set Caravan (Kratzert et al., 2023) are prominent examples. For water quality, such comprehensive data sets have been less common, but momentum is increasing with QUADICA (Ebeling et al., 2022) and the recently published CAMELS-Chem datasets from the US (Sterle et al., 2024) and from Switzerland (Do Nascimento et al., 2025), which include not only hydroclimatic drivers but also the temporal evolution of pollution sources (e.g., atmospheric nitrogen deposition and nitrogen surplus as diffuse sources). In parallel, a number of data sets now provide large samples of quality-controlled water quality time series (Zarei et al., 2025; Virro et al., 2021), further complemented by catchment or stream network characteristics (Fernandez et al., 2025; Minaudo et al., 2025).

Comprehensive LSH datasets have various applications. They support data-driven top-down approaches to identify trends and patterns in water quantity and quality time series, and when combined with contextual data help advance our understanding of underlying processes and hierarchies. They also provide forcing, calibration, and validation data for hydrological and water quality models (Nguyen et al., 2022; Van Meter and Basu, 2015). The increased availability of LSH datasets also propelled data-driven machine learning (ML) models using them for training, testing, and validation and improving their performance and generalization ability both in time and space (e.g. ungauged basins). ML models are widely applied and improved for discharge predictions (e.g., Kratzert et al., 2018; Heudorfer et al., 2025)

but also increasingly used for water quality parameters (Zhi et al., 2023; Zhi et al., 2021; Saha et al., 2023).

Here, we present the second version of QUADICA (water QUAlity, DIscharge and Catchment Attributes), a significant update to the original dataset (Ebeling et al., 2022). The first version of QUADICA has supported a wide variety of water quality studies, including the characterisation of catchments based on nutrient export processes across different spatial and temporal scales (Ebeling et al., 2021b; Ebeling et al., 2021a; Ehrhardt et al., 2021), effects of hydroclimatic extreme events on the catchments' nitrate export (droughts, Saavedra et al., 2024; floods, Saavedra et al., 2022), for nutrient stoichiometric characterisation (Wachholz et al., 2023), as well as for disentangling catchment processes using a process-based water quality model (e.g., Nguyen et al., 2022). A particular focus has been the linkage of observed instream water quality responses to drivers, enabled through the provided catchment attributes and driving forces in the form of diffuse nitrogen sources.

Recent shifts in environmental conditions, particularly hydrological extremes such as droughts, have substantial impacts on water quality (Saavedra et al., 2024; Winter et al., 2023; Dupas et al., 2025). This highlights the critical need to extend the QUADICA dataset to include more recent years covering extreme drought years and additional water quality and driver variables, thereby enhancing our ability to understand and address the evolving relationship between environmental change and water quality. Specifically, the update encompasses (1) longer time series up to 2020, capturing recent extreme events such as the 2018-2020 multi-year drought (e.g., Rakovec et al., 2022) with expected effects on solute export (e.g., Winter et al., 2023), (2) additional hydroecological time series such as oxygen and chlorophyll-a concentrations, enabling to move from water quantity and quality to ecological impact studies, (3) additional time series of driving forces including point sources and phosphorus inputs, allowing more comprehensive views on input-output (driver-response) relationships, useful e.g. for the quantification of nutrient legacies or model input data, and (4) larger amount of stations with joint water quantity and quality by linking to the recently published and widely known CAMELS-DE (Loritz et al., 2024) and Caravan-DE (Dolich et al., 2024) data sets. With this updated version, we aim to enhance the breadth of the large-sample water quality dataset QUADICA with additional depth, enabling us to address more research questions and ultimately support water quality management.

## 2 Station and catchment selection

The 1386 stations and corresponding delineated catchments from the original QUADICA data set (Ebeling et al., 2022) are retained in version 2. Although all stations lie within Germany, 17.9% of the catchments are transboundary with part of their area in a neighbouring country. Figure 1 shows the study area with updated information on the data availability. As for version 1, water quality and quantity data for QUADICA v2 were assembled from the German federal state authorities and merged with the data from QUADICA v1. This allowed us to extend the time series length as well as add new variables of water quality.

Similar to version 1, we assessed the data availability after quality control of the water quality time series data. After homogenization of variable names, units and formats across all federal states, the preprocessing steps included: (1) removal of duplicates and implausible values (i.e. zero and negative concentrations), (2) removal of outliers within each time series using a mean plus 4 standard deviations threshold (> 99.99 % confidence) in logarithmic space for concentrations and normal space for oxygen concentrations ($O_2$) and water temperature (T), (3) substitution of left-censored values using half of the detection limit, where applicable (i.e. nutrient and mineral concentrations). We additionally removed total organic carbon (TOC) concentrations >1000 mg $l^{-1}$, as we identified implausible plateaus of such high values in three stations, for which the outlier test failed.

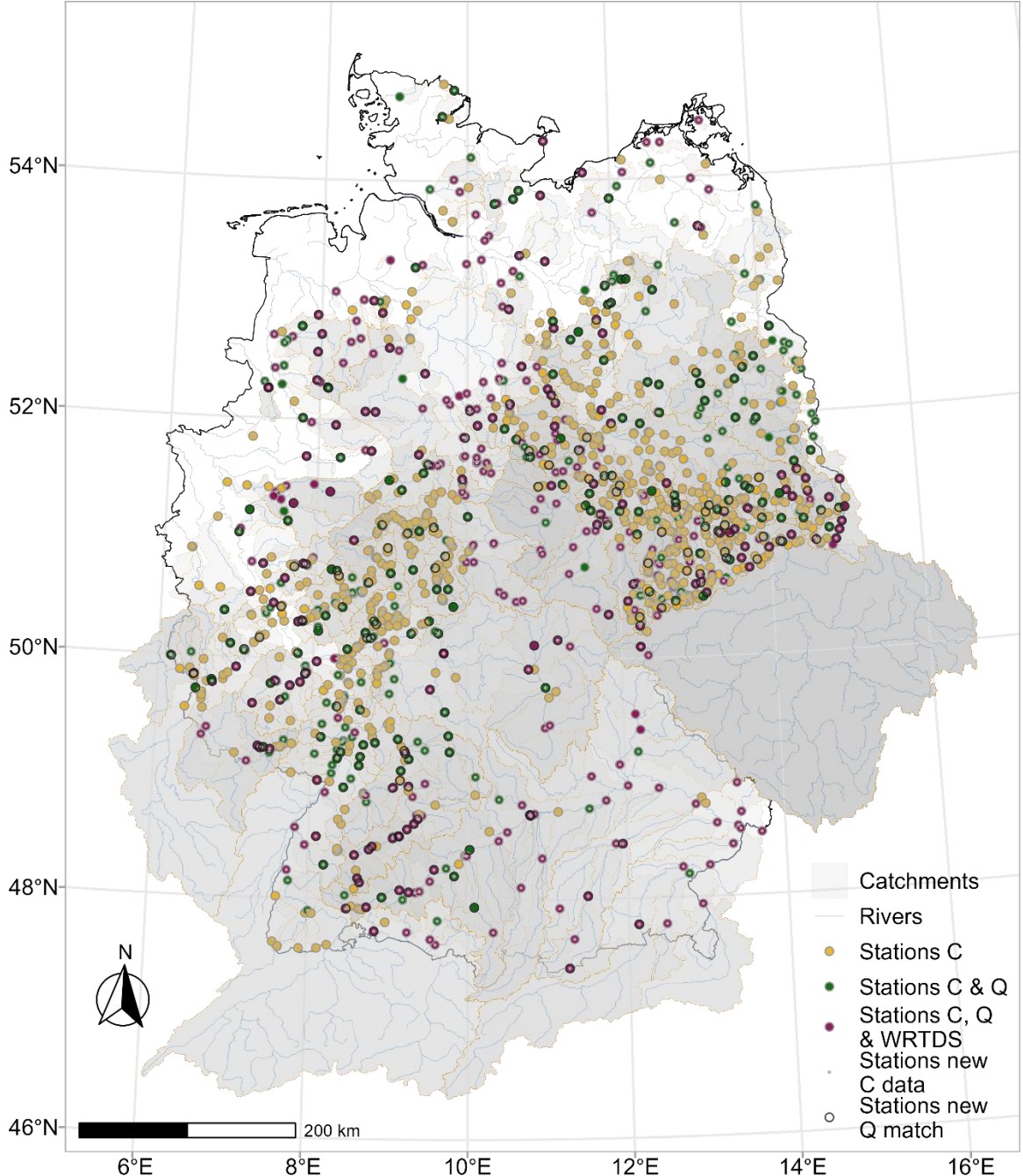

Figure 1: Stations and delineated catchments in relation to Germany (black line). Stations are colored according to their data availability, with C – concentration (water quality), Q – discharge (water quantity), and WRTDS - Weighted Regression on Time, Discharge and Season. Stations with extended water quality data (new C data) in version 2 are highlighted as well as stations with newly added continuous discharge data (new Q match) from matching with CAMELS-DE (Loritz et al., 2024) and Caravan-DE (Dolich et al., 2024) data sets (for details, refer to Section 3.2). The rivers displayed are taken from (De Jager and Vogt, 2007). WRTDS is available for stations with high data availability (see Section 3.1.2).

# 3 Time series

Time series data are provided for 1386 catchments (as in QUADICA v1) for water quality variables (Section 3.1) and water quantity (Section 3.2), and forcing variables both from meteorological drivers (Section 3.3) and nutrient (N and P) inputs from diffuse and point sources (Section 3.4). An overview of the provided (and newly added) variables is given in the following and in Table 1, while details are described in the following sections. Appendix B1 provides an overview of data files and respective metadata tables provided in the data repository. Note that due to limited data availability, not all water quality and quantity variables can be provided for all stations.

For water quality, QUADICA version 2 increases the number of variables by adding ammonium ($NH_4^+$-N) to the previously provided nutrient concentrations ($NO_3^-$-N, TN, $PO_4^{3-}$-P, TP, DOC, TOC), major ion concentrations ($SO_4^{2-}$, $Cl^-$, $Ca^{2+}$, $Mg^{2+}$), concentrations of $O_2$ and Chlorophyll-a (Chl-a), and water temperature (T). In version 2, dissolved inorganic nitrogen (DIN) was calculated as the sum of the preprocessed time series of inorganic nitrogen forms $NO_3^-$-N and $NH_4^+$-N, and, if available, $NO_2^-$-N. Note that, for simplicity, the charges are not always written in the following text. For water quantity, the number of stations with discharge data from daily observations was increased from 324 in version 1 to 637 in version 2. For nutrient inputs, time series of catchment-wise diffuse P inputs and point source inputs of N and P were added, while diffuse N sources were both updated as well as extracted from a European data source provided consistently with P.

**Table 1: Provided time series data, their basis (observed or estimated), aggregation type, temporal resolution and source of original data, which was used to calculate the aggregated data provided here. Bold font indicates the newly added variables in version 2 of the QUADICA data set. WRTDS -Weighted Regression on Time, Discharge and Season. Note that detailed metadata are provided for each data file in the repository, for an overview see Table B1.**

| Variable | Section | Data basis | Temporal (Spatial) Aggregation | Temporal resolution | File in repository | Source |
|---|---|---|---|---|---|---|
| Concentrations of nutrient species ($NO_3$-N, **$NH_4$-N**, **DIN**, TN, $PO_4$-P, | 3.1 | observed | median | annual | c_annual.csv | Musolff (2020); Ebeling et al. (2022) |
| | | daily estimated using WRTDS | median | monthly | wrtds_monthly.csv | Musolff (2020); Ebeling et al. (2022) |

| | | | | | | |
|---|---|---|---|---|---|---|
| TP, DOC, TOC), **major ions (SO₄, Cl, Ca, Mg), O₂** and **Chl-a,** and **T** | | observed | long-term median | monthly | c_q_avg_month s.csv | Musolff (2020); Ebeling et al. (2022) |
| Discharge | 3.2 | observed | median | annual | q_annual.csv | Musolff (2020); Ebeling et al. (2022); Loritz et al. (2024); Dolich et al. (2024) |
| | | observed | median | monthly | wrtds_monthly. csv | Musolff (2020); Ebeling et al. (2022); Loritz et al. (2024); Dolich et al. (2024) |
| | | observed | long-term median | monthly | c_q_avg_month s.csv | Musolff (2020); Ebeling et al. (2022); Loritz et al. (2024); Dolich et al. (2024) |
| Precipitation | 3.3 | observed gridded | sum (average) | monthly | climate_monthl y.csv | E-Obs (2018); (Cornes et al., 2018) |
| Potential evapotranspiration | 3.3 | estimated | sum (average) | monthly | climate_monthl y.csv | E-Obs (2018); (Cornes et al., 2018) |
| Mean air temperature | 3.3 | observed gridded | average (average) | monthly | climate_monthl y.csv | E-Obs (2018); (Cornes et al., 2018) |
| Diffuse N **(from two sources)** and **P input** as total | 3.4 | estimated | (average) | annual | input_N_P.csv | see Section 3.4 |
| Diffuse N input from agricultural areas | 3.4 | estimated | (average) | annual | input_N_P.csv | see Section 3.4 |
| **Point source N and P input** | 3.4 | estimated | (average) | annual | input_N_P.csv | see Section 3.4 |

147

## 3.1 Water quality time series

After quality control of the time series data, different temporal aggregation schemes were implemented

to provide consistent data sets. In QUADICA version 2, we provide the time series of annual medians

(Section 3.1.1), monthly medians for stations with high data availability (Section 3.1.2), and long-term
monthly averages (Section 3.1.3).

### 3.1.1 Annual median water quality variables

Annual median concentrations are provided based on the preprocessed time series (Section 2) for all
station-compound combinations. Along with the median concentrations, the number of samples
considered for the given value is provided as a control variable for users of the data set, allowing to subset
the data based on data availability.

The time series of annual median concentrations are visualized in Figures S1 and S2, while the
corresponding data density is shown in Figure 2 over the years as well as for the number of years covered
160 per station. A summary of data availability across all variables is provided in Table 2.

The highest data availability with more than 1370 stations covered is presented for the inorganic nitrogen
($NO_3$-N, $NH_4$-N, DIN) and phosphorus ($PO_4$-P) compounds, as well as for chloride (Cl), sulfate ($SO_4$),
oxygen ($O_2$) and water temperature (T). The highest temporal coverage stretches from the mid-2000s to
the mid-2010s. Overall, the median time series lengths vary between 13 (for Chl-a) and 24 ($O_2$, T) years.
The median number of samples per station varies between 104 (for Chl-a) and 205 (for T), while the
median average number of samples per year ranges from 10.1 (for DOC) to 11.9 (for $NO_3$-N, $PO_4$-P, and
167 T) and 12.0 (for Chl-a), i.e. corresponding to a monthly sampling frequency on average.

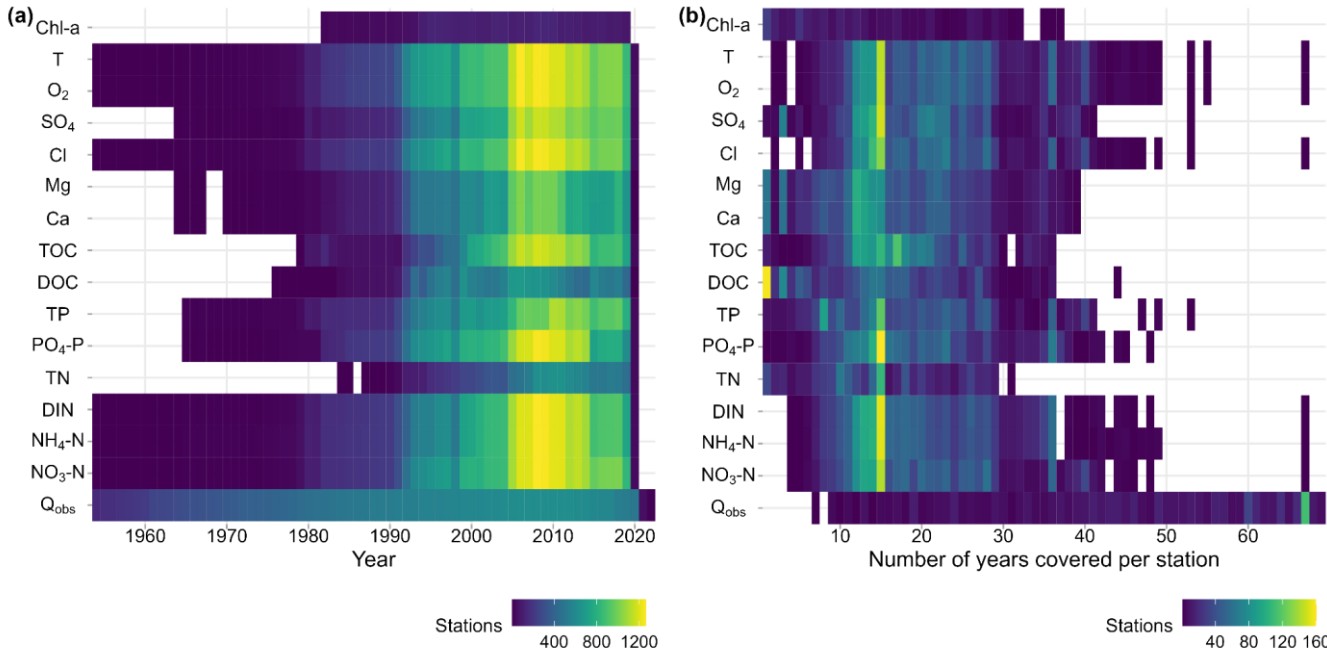

Figure 2: Temporal coverage of water quality and quantity time series data per compound: (a) number of stations with available annual medians per year and compound and (b) the number of years covered by each station per compound. For visualization purposes in (a) station counts from 1950 are shown, omitting one sample before 1954.

Table 2: Summary of stations and data availability for each water quality compound. The table provides the number of stations with the respective compound reported, the earliest and median start year of time series, median and maximum time series length in years across stations as well as the number of covered years (i.e. years with available data, with values provided in parenthesis), total number of grab samples (i.e. data points) for each compound, median number of grab samples per stations and median samples per year and station, number of outliers removed as the sum across all stations, and maximum fraction of outliers removed at one station. n - number, max. - maximum, * omitting one sample from 1900.

| Variable | NO$_3$-N | NH4-N | DIN | TN | PO$_4$-P | TP | DOC | TOC | Ca | Mg | Cl | SO4 | O$_2$ | T | Chl-a |
|---|---|---|---|---|---|---|---|---|---|---|---|---|---|---|---|
| Unit | mg l$^{-1}$ | mg l$^{-1}$ | mg l$^{-1}$ | mg l$^{-1}$ | mg l$^{-1}$ | mg l$^{-1}$ | mg l$^{-1}$ | mg l$^{-1}$ | mg l$^{-1}$ | mg l$^{-1}$ | mg l$^{-1}$ | mg l$^{-1}$ | mg l$^{-1}$ | °C | mg l$^{-1}$ |
| n stations | 1386 | 1386 | 1386 | 782 | 1379 | 1301 | 1167 | 1323 | 1337 | 1337 | 1380 | 1375 | 1379 | 1379 | 271 |
| Earliest start year | 1954* | 1954* | 1954* | 1984 | 1965 | 1965* | 1976 | 1979 | 1964 | 1964 | 1954 | 1964 | 1954 | 1954 | 1982 |
| Median start year | 1995 | 1997 | 1997 | 2005 | 1995 | 1996 | 1995 | 1999 | 1997 | 1997 | 1994 | 1997 | 1993 | 1993 | 1996 |
| Median time series length (years covered) | 22 (18) | 20 (17) | 20 (17) | 15 (14) | 21 (17) | 22 (17) | 19 (13) | 20 (17) | 19 (14) | 19 (15) | 23 (19) | 21 (17) | 24 (20) | 24 (20) | 13 (10) |
| Max. time series length in years (years covered) | 67* (67) | 67* (67) | 67* (67) | 31 (31) | 53 (48) | 53* (53) | 44 (44) | 37 (36) | 49 (39) | 49 (39) | 67 (67) | 53 (53) | 67 (67) | 67 (67) | 37 (37) |
| Total n samples (excl. outliers) | 375,990 | 364,301 | 356,262 | 139,948 | 350,507 | 323,520 | 171,123 | 291,898 | 232,926 | 232,412 | 372,123 | 299,412 | 462,508 | 396,836 | 65,632 |
| Median n samples per station | 194 | 190 | 190 | 168 | 183 | 177 | 130 | 179 | 145 | 144 | 191 | 181 | 203 | 205 | 104 |
| Median n samples per station and year | 11.9 | 11.8 | 11.8 | 11.4 | 11.9 | 11.7 | 10.1 | 11.7 | 11.1 | 11.0 | 11.8 | 11.8 | 11.8 | 11.9 | 12 |
| n outliers total | 88 | 292 | - | 74 | 212 | 506 | 339 | 950 | 119 | 228 | 666 | 212 | 219 | 8 | 50 |
| Max. fraction of outliers per station [%] | 1.9 | 3.4 | - | 2.2 | 5.8 | 2.9 | 3.2 | 7.2 | 2.4 | 3.8 | 2.3 | 4.0 | 2.1 | 1.1 | 2.6 |

## 3.1.2 Monthly median concentrations and mean fluxes for stations with high data availability

As in version 1 of QUADICA, we provide monthly and annually aggregated water quality data for the subset of stations with high data availability based on Weighted Regression on Time, Discharge and Season (WRTDS; Hirsch et al., 2010), referred to as 'WRTDS stations'. To fit WRTDS, we used the R package *EGRET* (version 3.0.9; Hirsch and De Cicco, 2015). WRTDS considers long-term trends,

seasonal components and discharge-dependent variability to estimate daily concentrations from low-frequency observations, e.g., from monthly grab samples (Hirsch et al., 2010). We included station and compound combinations using the same quality criteria as in QUADICA v1 on the preprocessed concentration data (Section 2). Accordingly, water quality time series had to cover at least 20 years, at least 150 samples, and no data gaps larger than 20 % of the total time series length. Discharge time series with daily temporal resolution are required to run WRTDS, but in contrast to version 1 of QUADICA, gaps in discharge were allowed with the consequence that no concentration estimate is provided for that day. The number of WRTDS stations varies between 97 for TN and 322 for Cl (Table 3), while the fraction of stations with high data availability varies between 12.0 % for TOC and 23.3 % for Cl.

As in QUADICA v1, monthly and annual values were only provided if 80% of the days of the respective period were covered. The provided water quality time series contain median concentrations, flow-normalized concentration, and mean flux estimates from WRTDS models. We now also added discharge-weighted mean concentrations. Discharge corresponds to the median observed, as WRTDS takes discharge as input and does not modify it (Section 3.2.2).

The model performance of WRTDS varies across water quality variables and stations with 64.1% of the station and compound combinations with $R^2 > 0.5$ and 58.2% with a percent bias <1% and 92.7% below <5%. Average performances per compound are given in Table 3, while the distribution of performance values is provided in Figure A3, as well as all individual values provided in the repository. The performance metrics should allow the users to select suitable catchments and compounds for reliable analysis.

Table 3: Number of stations with high data availability (WRTDS stations) for each compound and median coefficient of
determination of WRTDS models. The unit of all variables is mg l⁻¹.

| Variable | Number of WRTDS stations | Median $R^2$ | Median bias [%] |
|----------|--------------------------|--------------|-----------------|
| total | 347 | 0.58 | $-4.9*10^{-2}$ |
| $NO_3$-N | 317 | 0.64 | 0.20 |
| $NH_4$-N | 302 | 0.48 | 0.96 |
| DIN | 303 | 0.68 | 0.18 |
| TN | 97 | 0.71 | $5.1*10^{-3}$ |
| $PO_4$-P | 288 | 0.62 | -0.73 |
| TP | 270 | 0.48 | -0.53 |
| DOC | 140 | 0.45 | -0.65 |
| TOC | 195 | 0.46 | -0.40 |
| $Ca^{2+}$ | 175 | 0.62 | $2.8*10^{-2}$ |
| $Mg^{2+}$ | 174 | 0.57 | $-6.6*10^{-2}$ |
| Cl | 322 | 0.53 | $-3.9*10^{-2}$ |
| $SO_4$ | 234 | 0.67 | $5.5*10^{-2}$ |

## 211  3.1.3 Monthly long-term median concentrations

To be consistent with QUADICA v1, we provide monthly long-term medians, and 25th and 75th
percentiles (i.e. interquartile range), providing information on the average seasonality patterns of each
respective time series. Figure 3 shows the scaled medians indicating the variability of seasonal timing
across stations for each compound. For example, water temperature and oxygen show very similar
seasonality in terms of timing with summer maxima and summer minima, respectively, in contrast to,
e.g., $Ca^{2+}$, $Mg^{2+}$, DOC and TOC, for which seasonal timing varies strongly across stations. The nitrogen
and phosphorus species show dominant seasonal patterns, but still more variability across stations.

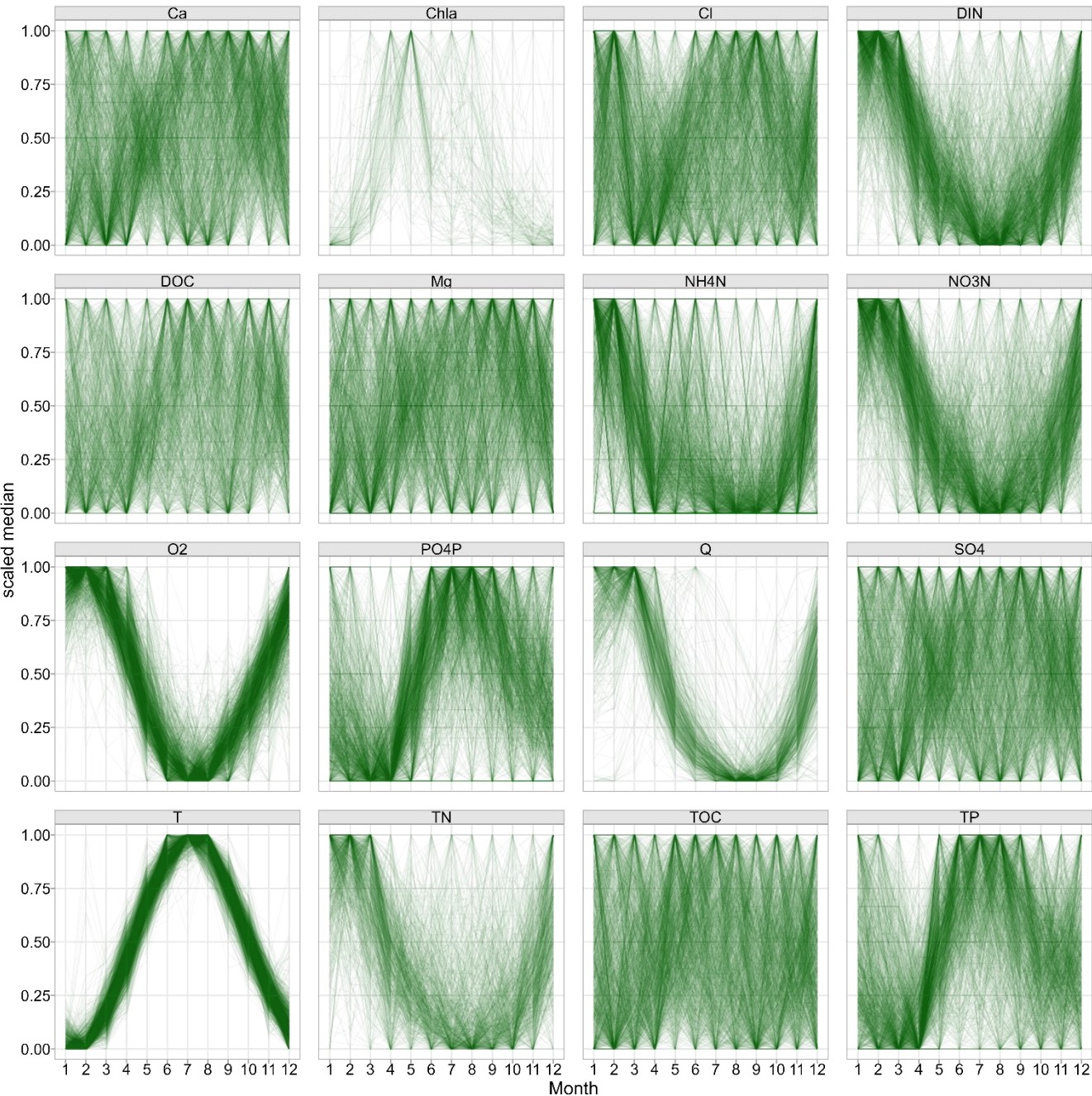

**Figure 3: Median monthly water quality observations inform about seasonal variability. Medians at each station are scaled to a range between 0 and 1. Note that only time series covering all 12 months are displayed.**

## 3.2 Water quantity time series

In total, discharge was provided for 637 stations, taking all data sources together. The earliest time series starts in 1893, the maximum number of stations with 620 stations with available discharge data was in 2011 and the longest time series extends until 2022.

From the QUADICA v1, we updated the discharge time series of 284 out of the 324 stations with daily data provided from our request to the authorities (232) and from GRDC (52) based on the matches identified in QUADICA v1. For the remaining stations, no updated data was provided.

In addition, we complemented the QUADICA discharge data from the CAMELS-DE (Loritz et al., 2024) and Caravan-DE (Dolich et al., 2024) data sets. We found 554 matches (449 from CAMELS, 105 from Caravan), out of which 313 stations had no matching discharge values in QUADICA yet, while 241 overlapped. We matched stations based on location and by manually checking if they lie on the same river. We differentiate cases between (1) close stations within a maximum distance of 1km (n=305) and (2) discharge stations that are further away. In the latter case, discharge stations could be located either (2i) upstream (n=202) or (2ii) downstream (n=47) of the water quality station. For (2), we accepted matches only if the relative difference between the intersected area of the CAMELS/Caravan and QUADICA catchments and the area of the QUADICA catchment was $\leq 30\%$. For downstream discharge stations (2ii), in addition, we accepted matches only if the CAMELS area was larger than the QUADICA area.

We additionally checked the correlations between QUADICA and CAMELS/Caravan time series with a median correlation coefficient of $r > 0.9999$ and only 5 out of the 241 overlapping stations with $r < 0.95$. We then used the discharge time series of the matched stations to fill up the QUADICA data. To account for differences in the locations (and thus catchments' area) of water quantity and water quality stations, we scaled the discharge of upstream discharge stations (i.e. case 2i) with the ratio between the QUADICA catchment area to the intersected area and of downstream stations (i.e. case 2ii) with the ratio between the QUADICA to CAMELS/Caravan catchment area. In case of several potential matches (because of identical station locations within CAMELS, n=24), we manually checked the time series to decide for the more complete one or merged them with priority on the more recent time series (n=2).

### 3.2.1 Annual median discharge

Similar to version 1, annual median discharge is aggregated from available observed discharge data. As described above (Section 3.2), daily Q data is available for 637 water quality stations. The data density distribution is visualised in Figure 2.

### 3.2.2 Monthly median discharge

Similar to version 1, monthly median discharge is provided for WRTDS stations. Note that we did not gap-fill the daily discharge time series for the WRTDS models, but instead provide median values only if at least 80% of the days are covered. This criterion refers both to the monthly and annual discharge data provided with the WRTDS data tables (as described in Section 3.1.2).

### 3.2.3 Monthly long-term median discharge

Similar to version 1 of QUADICA and the water quality variables (Section 3.1.3), long-term monthly median discharge, 25th and 75th percentiles, as well as the corresponding number of samples are provided. These values can be an indicator of average discharge seasonality across solutes and catchments in the long term.

## 3.3 Meteorological time series

As in QUADICA v1, meteorological time series (precipitation, potential evapotranspiration and average air temperature) are provided as spatial catchment averages on monthly resolution from 1950 to 2020. To obtain these, we followed the same approach on a newer version from the European Climate Assessment and Dataset project (E-Obs, 2018; Cornes et al., 2018) for the daily gridded data of climate variables. Moreover, for the stations for which we identified matches from the CAMELS-DE/Caravan-DE datasets the users can access daily time series of several hydrometeorological variables and different products therein (Dolich et al., 2024; Loritz et al., 2024). However, note that the water quality stations are not always located at the exact same location, please refer to Section 3.2 and the details provided in the data repository and data tables about the matches.

## 3.4 N and P input time series

### 3.4.1 Net N and P input from diffuse sources

Time series of catchment-scale N and P surplus (kg y$^{-1}$ ha$^{-1}$) from diffuse sources as shown in Figure 4 are provided (file: input_N_P.csv). The catchment-scale surplus corresponds to a soil surface budget and equals the balance between nutrient inputs minus the output on agricultural and non-agricultural areas at an annual resolution normalized to the catchment area. Inputs include mineral fertilizer, manure, other organic fertilizers (in the German N surplus dataset only; such as sewage sludge, compost and biogas digestate), atmospheric deposition, biological fixation (N surplus only), weathering (P surplus only) and seeds and planting material (in the German N surplus dataset only). Outputs correspond to crop and pasture removal.

For N surplus, two different data sets were used: 1. A Germany-wide county-scale data set as described in depth in QUADICA v1 (Ebeling et al., 2022; Behrendt et al., 2003; Häußermann et al., 2020), and 2. A European gridded data set (Batool et al., 2022).

For the first source of N surplus, the N surplus time series on agricultural areas were updated with the German data provided by Häußermann et al. (2020) for the period 1995-2021, following Ebeling et al. (2022). However, we refined the methodology to account for temporarily variant agricultural areas, following Sarrazin et al. (2022). The data now ranges from 1950-2021 (1950-2015 in the previous version). We extended the N surplus from non-agricultural areas until 2021 by calculating the sum of atmospheric deposition and biological N fixation as described in QUADICA v1. Note that the values for transnational catchments have higher uncertainties as they were calculated for the area within Germany only (for the corresponding fraction, see f_areaGer).

For the second source of N surplus, N surplus time series were extracted from a gridded, European-scale dataset (Batool et al., 2022) providing annual estimates of N surplus from 1850 to 2019 at 5 arcmin (~10 km at the equator) resolution. It covers both agricultural and non-agricultural soils. The N surplus time series across catchments from both sources are compared in Figure 4c, while a comparison of the datasets can be found in Batool et al. (2022). Overall, there is a correlation with r=0.72 across all catchments, which increases to r=0.76 when considering only the catchments with at least 70%, 95% or a 100% of

300 their catchment area within Germany. Additionally, differences can arise from methodological and scale
differences as well as uncertainties in general.

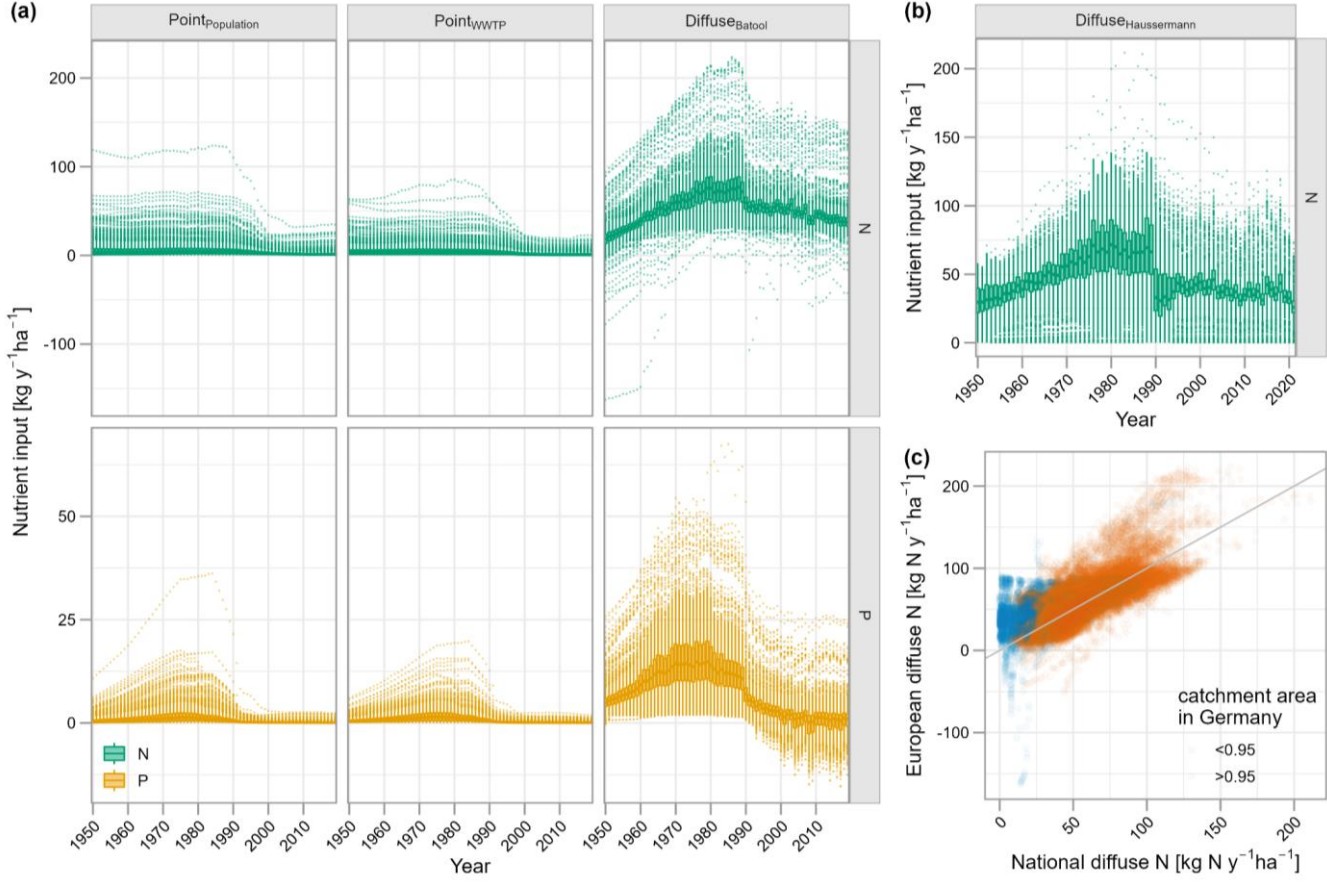

**Figure 4: Nitrogen and phosphorus input time series from different sources shown as distributions across all catchments. In (a) point sources data comes from Sarrazin et al. (2024)Sarrazin et al. (2024) corresponds to the ensemble mean from two different spatial disaggregation approaches based on population density (Point$_{Population}$) and WWTP data (Point$_{WWTP}$) (Section 3.4.2) and the ensemble mean of diffuse sources input of N from Batool et al. (2022) and of P from Batool et al. (2025) (Diffuse$_{Batool}$). In (b) diffuse source of N from Häußermann et al. (2020) is shown, while in (c) the diffuse N input values for each year and each catchment of the two data sets (from the German and European data basis) are compared, with the color indicating the fraction of catchment area within German boundaries (orange - ≥0.95, blue - <0.95). Note that: The boxes of the boxplots show the median, the 25th and 75th percentiles, while the whiskers extend up to 1.5*interquartile ranges with outliers beyond this range; Y axis scale is different for N and P.**

For P surplus, we used the European-scale dataset (Batool et al., 2025) constructed with the same spatial
and temporal resolution and a similar methodology as the one of N surplus. Both European datasets
quantify uncertainties in key components such as fertilizer use, manure allocation, and crop removal. For
QUADICA, we extracted the ensemble mean of the total N and P surplus estimates to assess diffuse

nutrient inputs relevant at the catchment scale. For further details on the data uncertainty, please refer to (Batool et al., 2022; Batool et al., 2025).

**3.4.2 N and P input from point sources from wastewater**

While in QUADICA v1, point source data are available for only one year (around 2016), QUADICA v2 provides time series of N and P point source inputs from wastewater for each catchment for the period 1950-2019. The data come from the gridded dataset of Sarrazin et al. (2024) for Germany. This data set provides estimates of N and P point sources, accounting for wastewater emissions that are treated in urban Wastewater Treatment Plants (WWTPs), including domestic and industrial (indirect) emissions, as well as untreated domestic emissions collected in the sewer system. These treated and untreated N and P emissions result from human excreta, with additional emissions for P due to the use of detergents. The data were constructed combining a modelling approach and observational data of WWTP N and P emissions. Sarrazin et al. (2024) provides ensemble runs from two methods to spatially disaggregate the data to grid resolution, that is, one based on population density and the other one based on recent WWTP outgoing N and P emissions. QUADICA v2 includes, for each catchment, two point source time series corresponding to the respective ensemble means of the two disaggregation approaches. For further details including time-dependent uncertainty of the two methods due to the shift in information detail and corresponding representativeness, please refer to Sarrazin et al. (2024).

# 4 Catchment attributes

The catchment attributes describe the topography, land cover, nutrient sources, lithology, and soils, and hydroclimate of the catchments. The attributes provided in QUADICA v1 were partly updated and complemented. New attributes include the Strahler order, updated land cover fractions from the CORINE Land cover dataset for 2018, the mean monthly Leaf Area Index (LAI), the soil pH in water and in $CaCl_2$-solution as well as updated average nutrient source and hydroclimatic characteristics. Here, we describe only updated and complemented characteristics; for a detailed description of the previous characteristics, please refer to QUADICA v1 (Ebeling et al., 2022). The metadata table of all characteristics in QUADICA

v2 is provided in Appendix B2 and Table S11 in the metadata of the data repository, while the attributes
data can be found in the file attributes.csv (see Appendix B1).

## 4.1 River network position

In the version 2 of QUADICA, we add the attribute of stream Strahler order, derived from the EU Hydro
data set (EEA, 2020). For each catchment, the largest Strahler order of streams intersecting the catchment
was selected and manually checked. The Strahler order provides context of the size and position of the
streams with headwater streams starting with Strahler order 1, going up to the order 8 for the downstream
part of the Elbe river. Most streams classify as order 3 (n=417) and 2 (n=321), i.e. small to medium sized
rivers.
To further support network analyses, we link each station to its next downstream station in the river
network and count the number of upstream stations, enabling spatially consistent analyses and modelling
of water quality patterns and network connectivity. More than half of the stations (731) have no station
further upstream, while 95 have no further downstream station.

## 4.2 Land cover

The fractions of land cover classes were calculated from the CORINE Land cover map (as in QUADICA
v1) but with the newer data set for 2018 (version 2020_20u1; EEA, 2019). We both provide level 1
(artificial, agricultural, forested land, wetland, and surface water cover) as well as level 2 data with refined
classes, as described in APPENDIX B.
For each catchment, the mean monthly LAI across the period 2003-2020 was extracted from high-quality
reprocessed MODIS LAI data (Yan et al., 2024). Generally, the LAI is defined as the ratio of green leaf
area to unit ground surface area, which can be estimated from spectral remote sensing data. The LAI
serves as an indicator for e.g. photosynthesis, evapotranspiration and rainfall interception capabilities of
vegetated areas.

## 4.3 Nutrient sources

Average inputs of nitrogen and phosphorus from diffuse and point sources for each catchment are provided based on the respective annual time series described in Section 3.4. We calculated the mean values starting from 1991 (i.e. 1991-2021 in case of Häußermann and 1991-2019 in case of Batool and Sarrazin), representing long-term average historic inputs since the year the Nitrate Directive was amended (EC, 1991). In addition, we calculated mean values over the last decade starting in 2010, representing current nutrient pollution pressures. We also renewed the measure of N source apportionment considering the data sets covering the same spatial scale for Germany, i.e. using the updated data product of the German-wide N surplus data and the newly added N point source data set for both the long-term period and the recent decade.

In addition, we provide catchment-averages of soil P budget data from the European data set provided by Panagos et al. (2022). The data set provides maps for P available for crops and P total in agricultural topsoil (0-20 cm) based on the Land Use and Cover Area frame Survey (LUCAS) as raster data with 500m resolution, as well as the soil P input and output budget components over the period 2011-2019. The input components inorganic fertilizers and manure are provided as vector data at NUTS (Nomenclature of Territorial Units for Statistics) 2 level, whereas the atmospheric deposition and chemical weathering data are in raster format. The extracted output components include the output through crop harvesting and removal of crop residues, both provided at NUTS2 level. Based on that we calculated the P surplus as a balance component at the soil level. For raster data we calculated the mean across each catchment, providing available and total P on agricultural soils, and scaled it to the catchment area by the fraction of agriculture based on CORINE land cover data (EEA, 2016). To estimate the catchment-scale values from the data sets at NUTS2 level, we first intersected them with the catchments, second calculated the fraction of agriculture to scale the input and output components, and finally calculated area-weighted means for each catchment.

## 4.4 Soil properties

In addition to average total soil nutrient content in the topsoil (0-20 cm), we added data on average soil pH. The topsoil pH in water and $CaCl_2$ 0.01 M solution was derived from the European soil chemistry

map, which is based on the LUCAS database (Ballabio et al., 2019). Historically, soil pH was often only measured in water. However, soil pH measured in a salt solution of $CaCl_2$ or KCl is now preferred, as it is less affected by electrolyte concentrations in the soil and thus provides a more consistent measurement of fluctuating salt content (Minasny et al., 2011). For comparability, the mean topsoil pH from both methods was extracted for each catchment.

## 4.5 Hydroclimatic characteristics

The hydrologic characteristics such as mean discharge and metrics of discharge variability were calculated from the updated observed daily discharge data for 637 stations (Section 3.2). We calculated long-term time series characteristics starting in November 1990 (hydrological year of 1991) until October 2020, i.e. covering 30 years if available. The exact starting and ending dates used for calculation are provided along with the characteristics, as well as information on missing values. For a list of characteristics, refer to Appendix B and the data repository. For those stations matching with CAMELS-DE/Caravan-DE (Dolich et al., 2024; Loritz et al., 2024), further hydrometeorological characteristics can be accessed directly from these datasets.

## 5 Limitations

Although some of the previously discussed limitations have been addressed, other limitations and uncertainties remain present in QUADICA v2.

We significantly increased the number of stations with discharge from daily time series and thus the number of stations with high data availability (WRTDS-stations) more than doubled to now 347 in total. Still, co-located water quantity and quality stations remain limited with less than half of the stations covered (637 out of 1386 stations).

Unfortunately, one of the main drawbacks related to data policies remains. More specifically, data handed over by federal state agencies cannot generally be handed over to third parties, so raw data of water quality and quantity cannot be provided here. We thus adhere to the provision of ready-to-use aggregated data, which can still serve various purposes, e.g. trend analysis (Ehrhardt et al., 2021) and long-term water quality modelling (Nguyen et al., 2022).

Uncertainties related to transboundary catchments (beyond the German borders) were reduced for the diffuse nutrient input time series by integrating the European data sets that have become available. However, the uncertainty for the point source time series, which only includes German territory, remains high and such stations may be excluded for certain analysis. For the diffuse N inputs, both time series from German as well as European data bases are provided enabling direct comparison to assess reliability and uncertainty related to the input time series.

# 6 Data availability

The data set can be accessed in the data repository under https://doi.org/10.4211/hs.c2866cd416b94ca386deb5758834311f (Ebeling et al., 2025). It includes all time series, catchment attributes and summary data as well as detailed data description files. Alongside with the repository, we provide an interactive R Shiny application that allows users to check data coverage and visualise selected time series. In addition, a browser-based web app is available for exploring the data set through the institutional UFZ GeoData Infrastructure, accessible at https://web.app.ufz.de/gdi/wq-monitor/en. Due to license agreements, the raw data itself cannot be published but are deposited in a long-term institutional repository (Musolff et al., 2020), for which metadata are deposited in a freely accessible repository (Musolff, 2020).

# 7 Conclusions

This paper aims to provide an updated and extended version of the QUADICA data set for Germany (Ebeling et al., 2022) to enhance both the breadth and the depth (Gupta et al., 2014). Therefore, we focused on describing the new additions in more detail. The main novelties are:

- Extension of water quality and quantity time series for four years up to 2020, covering severe drought years and generally longer time series (Section 3.1 and 3.2)
- New water quality parameters were added including those relevant for ecological impact studies such as oxygen, water temperature and chlorophyll-a concentrations (Section 3.1)

- Linkage to recently published large-sample water quantity data sets for Germany (CAMELS-DE by Loritz et al. (2024) and Caravan-DE by Dolich et al. (2024)) almost doubled the number of water quality stations with conjunctive continuous discharge data from 324 (version 1) to 637 (version 2), allowing for more comprehensive studies of water quantity and quality (Section 3.2)
- The increase in stations with daily discharge data has also increased the number of stations with high data availability (version 2: 347, before: 140) with monthly concentration time series derived from WRTDS models (Section 3.1.2)
- Addition of diffuse phosphorus input and nitrogen and phosphorus point source input time series for German catchments (Section 3.4)
- Addition and update of catchment characteristics including network position (Section 4)

These additions allow for further comprehensive investigations from drivers of nutrient pollution to water quality responses in streams, including ecological implications, and conjunctive water quality and quantity assessment.

# Appendix A

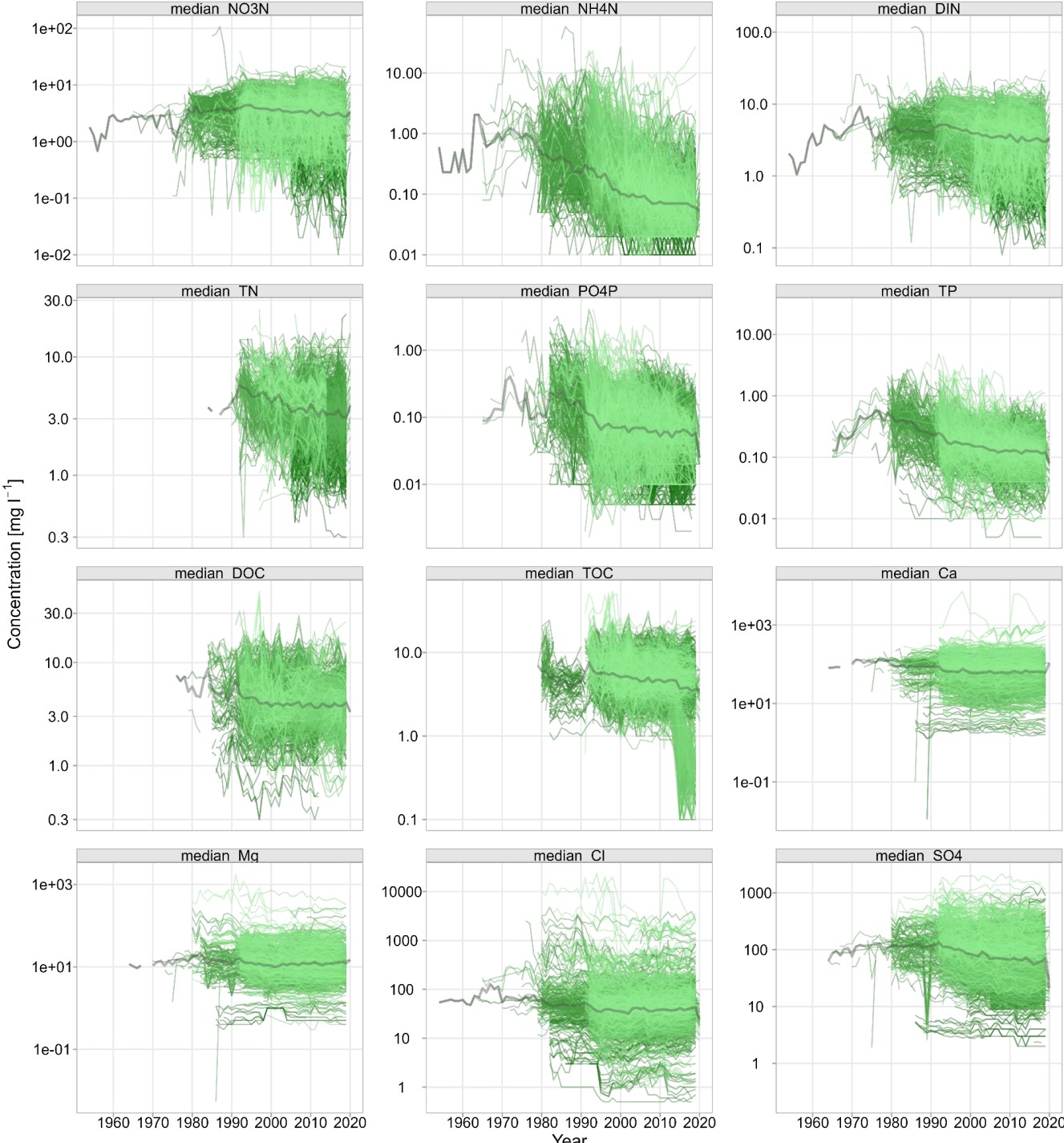

**Fig. A1: Annual median concentrations observed at the 1386 water quality stations (described in Table 1, Fig. 1 and Section 3.1).**
**The colors are gradual from light to dark corresponding to the OBJECTID numbers, the grey line shows the median concentration**
**across all annual medians.**

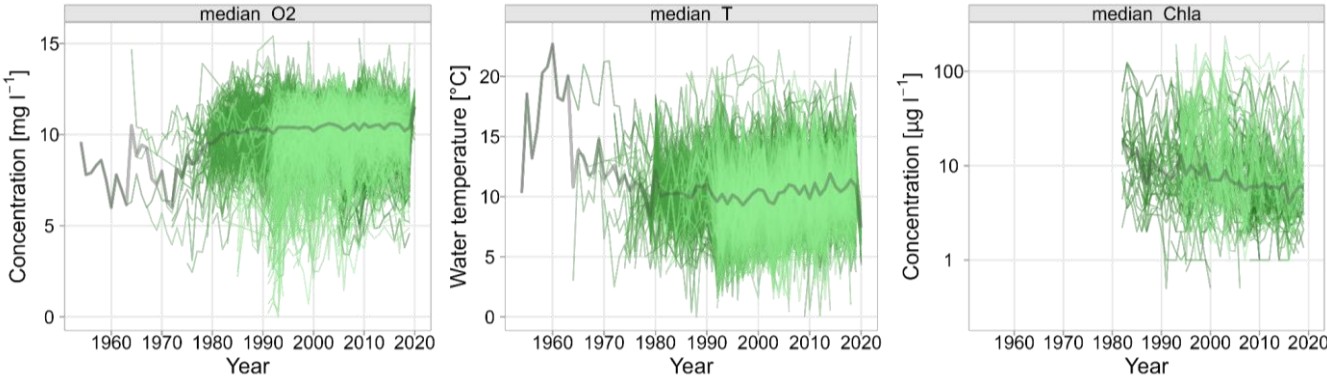

**Fig. A2: Annual median O₂ concentrations, water temperature, and chlorophyll-a concentration observed at the 1386 water quality**
**stations (described in Table 1, Fig. 1 and described in Section 3.1). The colors are gradual from light to dark corresponding to the**
**OBJECTID numbers.**

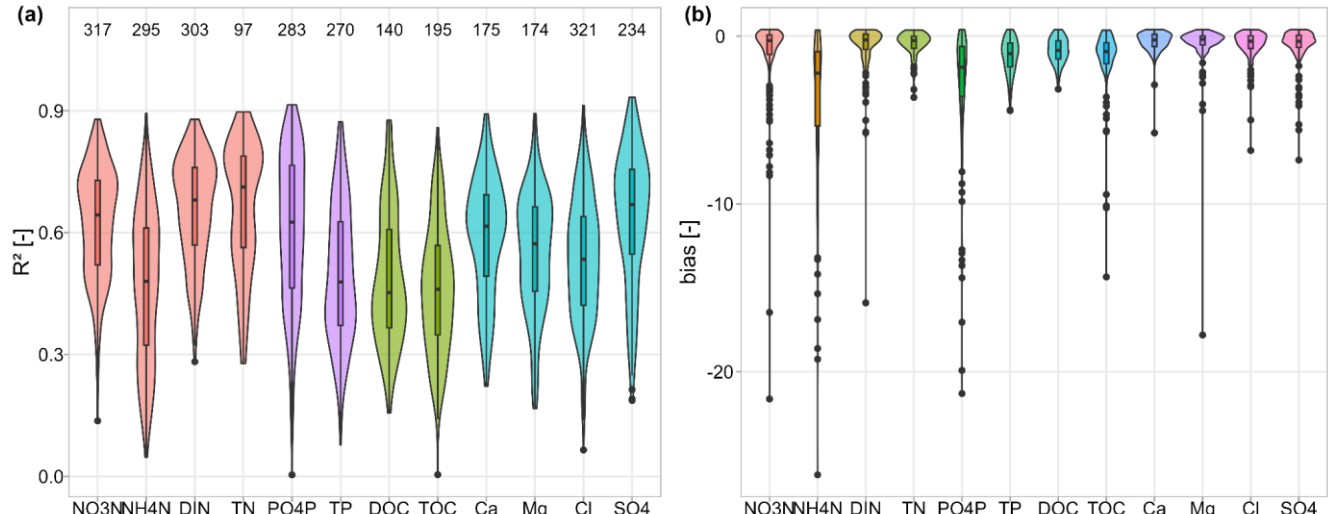

**Fig. A3: WRTDS-model performances for each compound: (a) coefficient of determination R² and (b) bias. Boxes highlight the**
**median and quartiles of each distribution. In (a) the number of time series is given on top for each compound. Colors according to**
**the substance group, i.e. nitrogen, phosphorus, organic carbon and major ions. Note that in (a) values of R2<0 were omitted,**
**accounting seven catchments for NH4-N, five for PO4-P, and one for Cl; in (b) values of bias < -30 were omitted, accounting five**
**values of NH4-N and one value for Cl. The users can define their quality criteria to subset the provided time series.**

# Appendix B

**Table B1: Overview of files and metadata tables in the description file (Metadata_QUADICA_v2.pdf) of the data repository.**

| Table in metadata file | Data file in repository | Corresponding section in manuscript |
| --- | --- | --- |
| S1 | metadata_c.csv | 3.1 general |
| S2 | metadata_q.csv | 3.2 general |
| S3 | wrtds_summary.csv | 3.1.2, 3.2.2 |
| S4 | c_annual.csv | 3.1.1 |
| S5 | c_q_avg_months.csv | 3.1.3, 3.2.3 |
| S6 | wrtds_monthly.csv, wrtds_annual.csv | 3.1.2, 3.2.2 |
| S7 | q_annual.csv | 3.2.1 |
| S8 | climate_monthly.csv | 3.3 |
| S9 | input_N_P.csv | 3.4 |
| S10 (same as Table B2) | attributes.csv | 4 |

**Table B2: Catchment attributes, associated methods and original data sources used for calculating the attributes. It contains both attributes from QUADICA v1 and the newly added and updated attributes. For more details see Section 4, data file: attributes.csv.**

| Category | Variable | Unit | Description and method | Data source |
| --- | --- | --- | --- | --- |
| General | OBJECTID | - | Unique identifier | |
| | Station | - | Station name | |
| | Area_km2 | km² | Catchment area | |
| | f_AreaGer | - | Fraction of catchment area within Germany | |
| Network | strahler_order | - | Strahler order based on EU Hydro river network | EEA (2020) |
| | id_downstream | - | OBJECTID of next downstream station | |
| | n_upstream | - | Number of upstream stations | |
| Topography | dem.mean | mamsl | Mean elevation of catchment, from DEM rescaled from 25 to 100 m resolution using average | EEA (2013) |
| | dem.median | mamsl | Median elevation of catchment, from DEM rescaled from 25 to 100 m resolution using average | EEA (2013) |

| | | | | |
|---|---|---|---|---|
| | slo.mean | ° | Mean topographic slope of catchment, from DEM | EEA (2013) |
| | slo.median | ° | Median topographic slope of catchment, from DEM | EEA (2013) |
| | twi.mean | - | Mean topographic wetness index (TWI, Beven & Kirkby, 1979) | EEA (2013) |
| | twi.med | - | Median topographic wetness index (TWI, Beven & Kirkby, 1979) | EEA (2013) |
| | twi.90p | - | 90[th] percentile of the TWI as a proxy for riparian wetlands (following Musolff et al., 2018) | EEA (2013) |
| | ddhad | $km^{-1}$ | Average drainage density of the catchment. Gridded drainage density is provided as the length of surface waters (rivers and lakes) per area from a 75km² circular area around each cell centered. | BMU (2000) |
| | DrainDens | $km^{-1}$ | Average drainage density of the catchment, calculated from EU-Hydro River Network and intersection with Catchment polygons (contains several implausible values (often too small values due to coarser resolution of river network)) | EEA (2016b) |
| Land cover | f_artif, f_artif_18 | - | Fraction of artificial land cover based on CORINE map from 2012 (f_artif) and 2018 (f_artif_18) | EEA (2016a), EEA (2019) |
| | f_agric, f_agric_18 | - | Fraction of agricultural land cover based on CORINE map from 2012 (f_agric) and 2018 (f_agric_18) | EEA (2016a), EEA (2019) |
| | f_forest, f_forest_18 | - | Fraction of forested land cover based on CORINE map from 2012 (f_forest) and 2018 (f_forest_18) | EEA (2016a), EEA (2019) |
| | f_wetl, f_wetl_18 | - | Fraction of wetland cover based on CORINE map from 2012 (f_wetl) and 2018 (f_wetl_18) | EEA (2016a), EEA (2019) |
| | f_water, f_water_18 | - | Fraction of surface water cover based on CORINE map from 2012 (f_water) and 2018 (f_water_18) | EEA (2016a), EEA (2019) |
| | f_urban, f_urban_18 | - | Fraction of Class 11 Level 2 CORINE Land Cover | EEA (2016a), EEA (2019) |

| | | | | |
|---|---|---|---|---|
| | f_industry, f_industry_18 | - | Fraction of Class 12 Level 2 CORINE Land Cover | EEA (2016a), EEA (2019) |
| | f_mine, f_mine_18 | - | Fraction of Class 13 Level 2 CORINE Land Cover | EEA (2016a), EEA (2019) |
| | f_urban_veg, f_urban_veg_18 | - | Fraction of Class 14 Level 2 CORINE Land Cover | EEA (2016a), EEA (2019) |
| | f_arable, f_arable_18 | - | Fraction of Class 21 Level 2 CORINE Land Cover | EEA (2016a), EEA (2019) |
| | f_agri_perm, f_agri_perm_18 | - | Fraction of Class 22 Level 2 CORINE Land Cover | EEA (2016a), EEA (2019) |
| | f_pastures, f_pastures_18 | - | Fraction of Class 23 Level 2 CORINE Land Cover | EEA (2016a), EEA (2019) |
| | f_agri_hetero, f_agri_hetero_18 | - | Fraction of Class 24 Level 2 CORINE Land Cover | EEA (2016a), EEA (2019) |
| | f_fores, f_fores_18 | - | Fraction of Class 31 Level 2 CORINE Land Cover | EEA (2016a), EEA (2019) |
| | f_scrub, f_scrub_18 | - | Fraction of Class 32 Level 2 CORINE Land Cover | EEA (2016a), EEA (2019) |
| | f_open, f_open_18 | - | Fraction of Class 33 Level 2 CORINE Land Cover | EEA (2016a), EEA (2019) |
| | lai_01, …, lai_12 | | Monthly mean leaf area index (LAI) as catchment average. The number indicates the month from 1 for January to 12 for December. | Yan et al. (2024) |
| | pdens | inhabitants $km^{-2}$ | Mean population density | *CIESIN (2017)* |
| Nutrient sources | Nsurp_Haussermann_from1991, Nsurp_Haussermann_from2010 | kg N $ha^{-1}$ $y^{-1}$ | Mean nitrogen (N) surplus per catchment from the German wide data set based on Häußermann et al. (2020) during the period 1991-2021 (from1991) and 2010-2021 (from2010). It includes the N surplus on agricultural and non-agricultural areas. Details in Section 3.4. | Bach et al. (2006); Bach and Frede (1998); Bartnicky and Benedictow (2017); Bartnicky and Fagerli (2006); Behrendt et al. (1999); Cleveland et al. (1999); Häußermann et al. (2020); Van Meter et al. (2017) |
| | Nsurp_Batool_from1991, Nsurp_Batool_from2010 | kg N $ha^{-1}$ $y^{-1}$ | Mean nitrogen (N) surplus per catchment from the European data set (Batool et al., 2022) during the period 1991-2021 (from1991) and 2010-2021 (from2010). It includes the N surplus on agricultural and non-agricultural areas. Details in Section 3.4. | Batool et al. 2022 |

| | | | |
|---|---|---|---|
| Psurp_Batool_from1991, Psurp_Batool_from2010 | kg N ha$^{-1}$ y$^{-1}$ | Mean phosphorus (P) surplus per catchment from the European data set (Batool et al., 2024) during the period 1991-2021 (from1991) and 2010-2021 (from2010). It includes the P surplus on agricultural and non-agricultural areas. Details in Section 3.4. | Batool et al. 2024 |
| Npoint_Pop_from1991, Npoint_Pop_from2010 | kg N ha$^{-1}$ y$^{-1}$ | Mean annual nitrogen (N) input from point sources with the population disaggregated approach during the period 1991-2021 (from1991) and 2010-2021 (from2010). | Sarrazin et al. 2024 |
| Ppoint_Pop_from1991, Ppoint_Pop_from2010 | kg N ha$^{-1}$ y$^{-1}$ | Mean annual phosphorus (P) input from point sources with the population disaggregated approach during the period 1991-2021 (from1991) and 2010-2021 (from2010). | Sarrazin et al. 2024 |
| Npoint_WWTP_from1991, Npoint_WWTP_from2010 | kg N ha$^{-1}$ y$^{-1}$ | Mean annual nitrogen (N) input with the wastewater treatment plant disaggregated approach during the period 1991-2021 (from1991) and 2010-2021 (from2010). | Sarrazin et al. 2024 |
| Ppoint_WWTP_from1991, Ppoint_WWTP_from2010 | kg N ha$^{-1}$ y$^{-1}$ | Mean annual phosphorus (P) input from point sources with the wastewater treatment plant disaggregated approach during the period 1991-2021 (from1991) and 2010-2021 (from2010). | Sarrazin et al. 2024 |
| f_Npoint_Pop_from1991, f_Npoint_Pop_from2010 | kg N ha$^{-1}$ y$^{-1}$ | Fraction of point source loads from total N input loads based on the population disaggregated point source data (Npoint_Pop) during the period 1991-2021 (from1991) and 2010-2021 (from2010). $f\_N_{point} = N_{point} / (N_{point} + Nsurp_{Haussermann})$ | |
| f_Npoint_WWTP_from1991, f_Npoint_WWTP_from2010 | kg N ha$^{-1}$ y$^{-1}$ | Fraction of point source loads from total N input loads based on the WWTP disaggregated point source data (Npoint_Pop) during the period 1991-2021 (from1991) and 2010-2021 (from2010). | |
| N_T_YKM2 | t N km$^{-2}$ y$^{-1}$ | Mean N input from point sources summing all N emission values provided in the EU domestic waste emissions data base | Vigiak et al. (2019); Vigiak et al. (2020) |
| P_T_YKM2 | t P km$^{-2}$ y$^{-1}$ | Mean P input from point sources summing all P emission values provided in the EU domestic waste emissions data base | Vigiak et al. (2019); Vigiak et al. (2020) |
| BOD_T_YKM2 | t O km$^{-2}$ y$^{-1}$ | Mean five-days biochemical oxygen demand (BOD) input from point sources summing all BOD emission values provided in the EU domestic waste emissions data base | Vigiak et al. (2019); Vigiak et al. (2020) |
| N_T_YEW | t N inh$^{-1}$ y$^{-1}$ | Calculated N input per person (from EU domestic waste emissions data base) | Vigiak et al. (2019); Vigiak et al. (2020) |

| | | | $N\_T\_YEW = N\_T\_YKM2 / nEW * Area\_km2$ | |
|---|---|---|---|---|
| | P_T_YEW | t P inh$^{-1}$ y$^{-1}$ | Calculated P input per person (from EU domestic waste emissions data base)<br>$P\_T\_YEW = P\_T\_YKM2 / nEW * Area\_km2$ | Vigiak et al. (2019); Vigiak et al. (2020) |
| | nEW | - | Calculated number of inhabitants,<br>nEW=pdens * Area_km2 | *CIESIN (2017)* |
| | n_UWWTP | - | Number of point sources from European data base (UWWTP data base) | EEA (2017) |
| | f_sarea | - | Fraction of source area in the catchment. Source areas were defined as seasonal, perennial cropland and grassland land cover classes using a highly resolved land use map (Pflugmacher et al., 2018) | Source areas based on Pflugmacher et al. (2018) |
| | het_h | m$^{-1}$ | Slope of relative frequency of source areas in classes of flow distances to stream as a proxy for horizontal source heterogeneity. For details refer to Ebeling, Kumar, et al. (2021) | Source areas based on Pflugmacher et al. (2018) |
| | R2_het_h | - | Coefficient of determination of horizontal source heterogeneity het_h | |
| | sdist_mean | m | Mean lateral flow distance of source areas to stream. For details refer to Ebeling, Kumar, et al. (2021) | Source areas based on Pflugmacher et al. (2018) |
| | het_v | - | Mean ratio between potential seepage and groundwater $NO_3$-N concentrations as proxy for vertical concentration heterogeneity. For details refer to Ebeling, Kumar, et al. (2021) | Knoll et al. (2020) |
| | P_available_agri | kg ha-1 | Available P stock in the agricultural topsoil (0-20 cm) | Panagos et al. (2022) |
| | P_available | | Available P stock from agricultural topsoil scaled to the whole catchment area, i.e. P_available_agri is scaled by the fraction of agriculture (f_agric) | Panagos et al. (2022), EEA (2016) |
| Lithology and soils | f_calc | - | Fraction of calcareous rocks (Lithology level 4) | BGR & UNESCO (eds.) (2014) |
| | f_calc_sed | - | Fraction of calcareous rocks and sediments (Lithology level 4, coarse and fine sediments aggregated) | BGR & UNESCO (eds.) (2014) |
| | f_magma | - | Fraction of magmatic rocks (Lithology level 4) | BGR & UNESCO (eds.) (2014) |
| | f_metam | - | Fraction of metamorphic rocks (Lithology level 4) | BGR & UNESCO (eds.) (2014) |

| | | | |
|---|---|---|---|
| f_sedim | - | Fraction of sedimentary aquifer (Lithology level 4, coarse and fine sediments aggregated) | BGR & UNESCO (eds.) (2014) |
| f_silic | - | Fraction of siliciclastic rocks (Lithology level 4) | BGR & UNESCO (eds.) (2014) |
| f_sili_sed | - | Fraction of siliciclastic rocks and sediments (Lithology level 4, coarse and fine sediments aggregated) | BGR & UNESCO (eds.) (2014) |
| f_consol | - | Fraction of consolidated rocks (Lithology Level 5) | BGR & UNESCO (eds.) (2014) |
| f_part_consol | - | Fraction of partly consolidated rocks (Lithology Level 5) | BGR & UNESCO (eds.) (2014) |
| f_unconsol | - | Fraction of unconsolidated rocks (Lithology Level 5) | BGR & UNESCO (eds.) (2014) |
| f_porous | - | Fraction of porous aquifer (code 1 and 2 of aquifer type) | BGR & UNESCO (eds.) (2014) |
| f_porous1 | - | Fraction of porous aquifer (code 1 of aquifer type) | BGR & UNESCO (eds.) (2014) |
| f_porous2 | - | Fraction of porous aquifer (code 2 of aquifer type) | BGR & UNESCO (eds.) (2014) |
| f_fissured | - | Fraction of fissured aquifer (code 3 and 4 of aquifer type) | BGR & UNESCO (eds.) (2014) |
| f_fiss1 | - | Fraction of fissured aquifer (code 3 of aquifer type) | BGR & UNESCO (eds.) (2014) |
| f_fiss2 | - | Fraction of fissured aquifer (code 4 of aquifer type) | BGR & UNESCO (eds.) (2014) |
| f_hard | - | Fraction of locally aquiferous and non-aquiferous aquifer (code 5 and 6 of aquifer type) | BGR & UNESCO (eds.) (2014) |
| f_hard1 | - | Fraction of locally aquiferous rocks (code 5 of aquifer type) | BGR & UNESCO (eds.) (2014) |
| f_hard2 | - | Fraction of non-aquiferous rocks (code 6 of aquifer type) | BGR & UNESCO (eds.) (2014) |
| f_inwater | - | Fraction of inland water (code 200 of aquifer type) | BGR & UNESCO (eds.) (2014) |
| f_ice | - | Fraction of snow or ice field (code 300 of aquifer type) | BGR & UNESCO (eds.) (2014) |
| dtb.median | cm | Median depth to bedrock in the catchment | Shangguan et al. (2017) |
| f_gwsoils | - | Fraction of water-impacted soils in the catchment (from soil map 1:250,000), including | BGR (2018) |

| | | | stagnosols, semi-terrestrial, semi-subhydric, subhydric and moor soils | |
|---|---|---|---|---|
| | f_sand f_silt f_clay | - | Mean fraction of sand in soil horizons of the top 100 cm Mean fraction of silt in soil horizons of the top 100 cm Mean fraction of clay in soil horizons of the top 100 cm | FAO/IIASA/ISRIC/ISSCAS/JRC (2012) |
| | f_clay_agri | | Mean fraction of clay in soil horizons of the top 100 cm on agricultural land use (Class 2 Level 1 CORINE; see f_clay and f_agric) | FAO/IIASA/ISRIC/ISSCAS/JRC (2012), EEA (2016a) |
| | WaterRoots | mm | Mean available water content in the root zone from pedo-transfer functions | Livneh et al. (2015); Samaniego et al. (2010); Zink et al. (2017) |
| | thetaS | - | Mean porosity in catchment from pedo-transfer functions | Livneh et al. (2015); Samaniego et al. (2010); Zink et al. (2017) |
| | soilN.mean | g kg$^{-1}$ | Mean top soil N in catchment | Ballabio et al. (2019) |
| | soilP.mean | mg kg$^{-1}$ | Mean top soil P in catchment | Ballabio et al. (2019) |
| | soilCN.mean | - | Mean top soil C/N ratio in catchment | Ballabio et al. (2019) |
| | soilpH_CaCl | - | Mean top soil pH from CaCl2 0.01 M solution in the catchment | Ballabio et al. (2019) |
| | soilpH_H2O | - | Mean top soil pH measured in water in the catchment | Ballabio et al. (2019) |
| Hydrology | Q_StartDate | YYYY-MM-DD | Starting date of Q time series used for calculating hydrological indices (from November 1990, if possible and at least 3 years of data (all 637 stations fulfilled that)) | |
| | Q_EndDate | YYYY-MM-DD | End date of Q time series used for calculating hydrological indices (up to October 2020 if available) | |
| | Q_gaps | boolean | If there are missing discharge values (a gap) in between Q_StartDate and Q_EndDate, the value is 1; without any gap the value is 0. | |
| | Q_nNAs | - | Number of missing values in between Q_StartDate and Q_EndDate. | |
| | Q_mean | m³ s$^{-1}$ | Mean discharge (data for the period Q_StartDate-Q_EndDate) | |
| | Q_median | m³ s$^{-1}$ | Median discharge (data for the period Q_StartDate-Q_EndDate) | |

| | | | | |
|---|---|---|---|---|
| | Q_spec | mm y$^{-1}$ | Mean annual specific discharge (data for the period Q_StartDate-Q_EndDate) | |
| | Q_CVQ | - | Coefficient of variation of time series of daily Q (data for the period Q_StartDate-Q_EndDate) | |
| | Q_medSum | m³ s$^{-1}$ | Median summer discharge (months May-October) (data for the period Q_StartDate-Q_EndDate) | |
| | Q_medWin | m³ s$^{-1}$ | Median winter discharge (months November-April) (data for the period Q_StartDate-Q_EndDate) | |
| | Q_Sum2Win | - | Seasonality index of Q, as ratio between median summer and median winter Q (data for the period Q_StartDate-Q_EndDate) | |
| | BFI | - | Base flow index calculated according to WMO [2008] with *lfstat* package (version 0.9.4) in R (data for the period Q_StartDate-Q_EndDate) | |
| | flashi | - | Flashiness index of Q as the ratio between 5 % percentile and 95 % percentile of Q time series (data for the period Q_StartDate-Q_EndDate) | |
| Climate | P_mm | mm y$^{-1}$ | Mean annual precipitation (period 1986-2015) | Cornes et al. (2018) |
| | P_SIsw | - | Seasonality of precipitation as the ratio between mean summer (Jun-Aug) and winter (Dec-Feb) precipitation (period 1986-2015) | Cornes et al. (2018) |
| | P_SI | - | Seasonality index of precipitation as the mean difference between monthly averages of daily precipitation and year average of daily precipitation (period 1986-2015) | Cornes et al. (2018) |
| | P_lambda | d$^{-1}$ | Mean precipitation frequency λ as used by Botter et al. (2013) with rain days for precipitation above 1 mm (period 1986-2015) | Cornes et al. (2018) |
| | P_alpha | mm d$^{-1}$ | Mean precipitation depth as used by Botter et al. (2013) with rain days for precipitation above 1 mm (period 1986-2015) | |
| | PET_mm | mm y$^{-1}$ | Mean annual potential evapotranspiration (period 1986-2015) | Cornes et al. (2018) |
| | AI | - | Aridity index as AI=PET_mm/P_mm (period 1986-2015) | Cornes et al. (2018) |
| | T_mean | °C | Mean annual air temperature (period 1986-2015) | Cornes et al. (2018) |

**Author contributions.**

The study was conceptualized by PE, AM, and RK. PE played a key role in data management, ensuring the quality, homogenization, and preprocessing of the data, as well as developing the methodology for matching and merging CAMELS/Caravan discharge data. PE also prepared the results, created visualizations, wrote the first draft of the manuscript and revised the manuscript. AW, US collected the water quality and quantity data from federal authorities and together with AH contributed to data quality control. SH, TN contributed to matching and merging QUADICA-CAMELS and Caravan stations, SH additionally extracted some new catchment attributes. Additionally, TN developed a Shiny App to facilitate data exploration in the data repository, with additions from PE. MB, FS, RK provided the catchment N and P input data. RK also contributed the climate and LAI data.

**Competing interests.** The authors declare that they have no conflict of interest.

**Acknowledgements.** We gratefully thank all data collectors, processors and providers including the federal state environmental agencies and all other contributors to this data set. We thank Nils Turner for his contributions to water quality data control, José Ledesma for discussions on the quality of discharge data, Sabine Attinger and Jan H. Fleckenstein for their initial input to QUADICA v1, Linus Schauer for providing the Strahler order as catchment descriptor, and Nicoletta Leitgeb for providing up- and downstream stations. We gratefully acknowledge Martin Bach and Uwe Häußermann, Justus-Liebig-University of Giessen, for the provision of the two data sets on the agricultural N surplus data for Germany. We acknowledge the E-OBS data set from the EU-FP6 project UERRA (http://www.uerra.eu) and the Copernicus Climate Change Service, and the data providers in the ECA&D project (https://www.ecad.eu). The authors additionally acknowledge several organizations for the data products used here, including the BfG, BGR, SGD, EEA, FAO, IIASA, ISRIC, ISSCAS, and JRC. Large Language Models (LLM), in particular Llama3 405 embedded in the Helmholtz AI Jülich service Blablador, have been used to increase readability of parts of the text - we thank the providers.

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
