# Peer review of "QUADICA v2: Extending the large-sample data set for water"

_Earth System Science Data, 2025_

## Author Response (AR1)

**Reply to comments from Referee 1 of the preprint in ESSD "QUADICA v2: Extending the large-sample data set for water QUAlity, DIscharge and Catchment Attributes in Germany" by Ebeling et al.**

Ebeling and colleagues updated the very relevant QUADICA dataset in this extension of the original with both more measurements and data. It is undeniable the relevance of the current contribution, and given some concerns are clarified I see no further obstacle in eventually having the contribution published in ESSD.

*We thank Referee 1 for the positive assessment of our work and the helpful suggestions below. We address the individual comments with responses in blue for clarity.*

1. L31: Is the data updated until 2020 or 2022? I see some parts of the paper where the 2020 is mentioned, and others where there is the 2020. I recommend to be coherent throughout the manuscript with one end_date.

*Thank you for this careful spotting. The water quality data in QUADICA in fact extends until 2020, while the discharge data at a few stations extends until 2022. It is possible to exclude the years 2021 and 2022 for a consistent end year, however, we prefer to keep all data as the time series length and covered period is quite individual for the different stations and variables anyway. Even the station-compound combinations that extend until 2020 are much smaller (374) than those ending in 2019 (9683). Thus, we prefer to provide all data as the user has to select the time span that is sufficiently covered and relevant. To avoid confusion, we modify the sentence slightly:*

*"Specifically, QUADICA v2 extends the water quality time series of the first version up to 2020 and introduces new variables, including water temperature, oxygen, and chlorophyll-a concentrations, as well as concentrations of ammonium, sulfate, and geogenic solutes like calcium."*

2. L36-37: Impressive and very useful. However as a reader I found it difficult to understand what the authors meant at first. I would please ask the authors to rephrase this part to make it more clear to the reader at first glance their significant contribution.

*We will rephrase the sentence to increase its clarity to the reader.*

*"Furthermore, we effectively doubled the number of stations with combined water quality and quantity data – now covering 637 out of the total 1386 stations – by integrating QUADICA with the hydrological large-sample datasets CAMELS-DE and Caravan-DE."*

3. L57: Please consider using the published version here in the reference instead of the preprint (https://doi.org/10.1038/s41597-025-05625-1)

do Nascimento, T.V.M., Höge, M., Schönenberger, U. et al. Swiss data quality: augmenting CAMELS-CH with isotopes, water quality, agricultural and atmospheric data. Scientific Data 12, 1283 (2025). https://doi.org/10.1038/s41597-025-05625-1

Thank you for the hint, we will update the reference.

4. Please consider adding the recent and also relevant contribution by Zarei and colleagues covering Iranian rivers (in case it makes sense for the authors):

Zarei, E., Noori, R., Jun, C. et al. A Comprehensive Water Chemistry Dataset for Iranian Rivers. Scientific Data 12, 1646 (2025). https://doi.org/10.1038/s41597-025-05932-7

Thank you, we included the reference and also a few others providing valuable water quality time series data.

5. L017-111: This is a very long parenthesis. Could you please rephrase to improve the flow in the text?

We rephrased the sentence as follows:

"After homogenization of variable names, units and formats across all federal states, the preprocessing steps included:

(1) removal of duplicates and implausible values (i.e. zero and negative concentrations),

(2) removal of outliers within each time series using a mean plus 4 standard deviations threshold (> 99.99 % confidence) in logarithmic space for concentrations and normal space for oxygen concentrations (O2) and water temperature (T),

(3) substitution of left-censored values using half of the detection limit, where applicable (i.e. nutrient and mineral concentrations)."

6. Maybe I am being too pendant about the map, but would not make sense to also add the north and one scale bar in km also? since it is a map in the end.

We added north arrow and scale bar to Figure 1. Thanks for the detailed revision.

7. Data records: I really appreciate that the authors already included a descriptive metadata in the dataset. It is great that you point users to the section in the manuscript that each table refers. However, I would appreciate if you could also insert such link in the manuscript. So for ex, in section 3.1.1 you could already point that the data is further described in Table S2? I assume that the metadata works as a Supporting information? Or in case not, I would strongly recommend to have a Supporting information with the same information as in the metadata downloadable with the dataset!

We are happy that the reviewer appreciates the metadata provided along the data set. Thank you for the suggestion to more clearly link it in the manuscript.

According to ESSD guidelines data sets should be published in repositories and we consider the metadata as a direct part of that data set (exert from ESSD guidelines: "In general, supplementary material that can be hosted in alternative sites such as FAIR-aligned data repositories should be placed there.")

Therefore, we added a table to the manuscript Appendix (new Table B1) providing an overview of metadata tables of the individual data files provided in the repository. We also added a column to Table 1 providing the file names of the corresponding time series data, and several hints in the text to respective metadata Tables.

To further support the users of the repository with easy overview of the information, we added a similar Table (List of Tables) in the beginning of the metadata file in the repository, as the number of Tables S1-S10 is large and tables are partly spacious.

8. L243-244: What do the authors mean here?

We decided to delete the sentence to avoid confusion as this information is not essential for QUADICA version 2 users. To still clarify: we were referring to the fact that in the QUADICA version 1, we had provided discharge information at stations where we had information on discharge during the grab sampling date of the concentration measurements but no continuous discharge time series. For version 2, we did not include those additional discharge values as we did not have any update here and also because the overall number of discharge stations from continuous discharge data was increased significantly.

9. L328: I could not find the catchment attributes data in the dataset. The metadata points to another repository with last updated data from 2022. Is this the correct path? If yes, is there a reason why you are not publishing all the QUADICA v02 dataset together?

We are very sorry for this confusion as we had included the old link of the first version in our manuscript. The catchment attributes are indeed provided along with all the other data as attributes.csv in the second version. Here is the link to version 2 of QUADICA for clarity:

https://www.hydroshare.org/resource/c2866cd416b94ca386deb5758834311f/

10. Section 3.4: In the metadata you point to N_SURPLUS data, but in the manuscript section I had the impression that you had available both N and P surplus. Is it true? if yes, where is it stored? Again, I think that having a table, like S7 directly mentioned in the manuscript would help the users and readers!

Again, we are very sorry for this confusion, the nutrient input data is provided in file input_N_P.csv and includes both N and P input as described in the manuscript. We added a reference to the respective file to the text. Here is the link to version 2 of QUADICA for clarity:

https://www.hydroshare.org/resource/c2866cd416b94ca386deb5758834311f/

11. 4.2 Why is LAI only from 2003-2018 made available? Is there any reason for not including the years up to 2020?

Thank you for bringing up this topic. Indeed, for consistency, we updated the data to cover until 2020. Although we think that the difference is not large for long-term averaged monthly LAI values provided. For more detailed data, the data set users can extract any kind of geodata using the polygons provided as catchment boundaries.

Given these clarifications are made, I would be willing to further review the paper for next steps.

Thank you, very much for your willingness to review our paper in next steps.

**Reply to comments from Referee 2 of the preprint in ESSD "QUADICA v2: Extending the large-sample data set for water QUAlity, DIscharge and Catchment Attributes in Germany" by Ebeling et al.**

In this data paper, Ebeling et al. present an updated version of their QUADICA database, which provides data on water quality and its controlling factor across Germany. Compared to the original version, QUADICA v2 contains additional water quality parameters, monitoring stations and data on 'driving forces'.

It is a comprehensive and well-described database, and the updates are significant enough to justify a second data paper on QUADICA. The database will be invaluable to scientists working on water quality and its controlling factors.

We thank the reviewer for its positive assessment of our work and value of the data set.

The only drawback of QUADICA v2 is that it only includes aggregated annual or monthly data, not raw data. This is a pity, as it will prevent its use for some potential statistical analyses. However, the authors explain that they had no choice because data providers have strict policies regarding data sharing.

We agree with the reviewer that this is a drawback and acknowledge the reviewers understanding given the data policies.